# Research Progress on Nanoparticles-Based CRISPR/Cas9 System for Targeted Therapy of Tumors

**DOI:** 10.3390/biom12091239

**Published:** 2022-09-05

**Authors:** Dengyun Nie, Ting Guo, Miao Yue, Wenya Li, Xinyu Zong, Yinxing Zhu, Junxing Huang, Mei Lin

**Affiliations:** Taizhou People’s Hospital Affiliated to Nanjing University of Chinese Medicine, Taizhou 225300, China

**Keywords:** CRISPR/Cas9, nanoparticles, cancer, delivery, therapy

## Abstract

Cancer is a genetic mutation disease that seriously endangers the health and life of all human beings. As one of the most amazing academic achievements in the past decade, CRISPR/Cas9 technology has been sought after by many researchers due to its powerful gene editing capability. CRISPR/Cas9 technology shows great potential in oncology, and has become one of the most promising technologies for cancer genome-editing therapeutics. However, its efficiency and the safety issues of in vivo gene editing severely limit its widespread application. Therefore, developing a suitable delivery method for the CRISPR/Cas9 system is an urgent problem to be solved at present. Rapid advances in nanomedicine suggest nanoparticles could be a viable option. In this review, we summarize the latest research on the potential use of nanoparticle-based CRISPR/Cas9 systems in cancer therapeutics, in order to further their clinical application. We hope that this review will provide a novel insight into the CRISPR/Cas9 system and offer guidance for nanocarrier designs that will enable its use in cancer clinical applications.

## 1. Introduction

Cancer is a major risk to the health and lives of people all over the world [1]. However, remarkable progress has been made in the prevention and treatment of cancers with in-depth understanding of tumor biology. The genetic landscape of cancer has become a consensus. Human cells carry both oncogenes and tumor suppressor genes. Under normal circumstances, these two genes antagonize each other to keep the balance of coordination as well as enabling precise regulation of cell growth, proliferation and decline [2]. Various alterations in DNA sequence of these two kinds of genes, such as mutation, deletion and deformity, underlie the development of tumors [3]. Massive sequencing projects have revealed that these genetic alterations are either specific to a certain type of cancer [4,5] or common to many cancer entities [6,7]. These sequence variants can be transmitted genetically, causing susceptibility to cancer, or can arise from somatic mutations. Thus, genome editing technology is a promising therapeutic tool for cancer therapy.

The systematic functional analysis of genes and mutations have been slow and arduous. In recent years, the rapid development of CRISPR/Cas9 technology has greatly accelerated the process of genome engineering. Since its first use as a genome editing tool in 2013 in mammalian cells [8,9], the toolbox of CRISPR/Cas9 has continued to expand to not only modify the genome sequence of cells and organisms [10], but also to introduce epigenetic and transcriptional modifications [11]. This implies CRISPR/Cas9 technology can also be used in targeted gene therapy of tumors. However, the CRISPR/Cas9 system depends on a delivery vector to play a significant role in gene therapy of tumors due to its special nature, which currently greatly limits its wide application in cancer treatment. Firstly, the CRISPR/Cas9 system is extremely unstable and can be digested by endonucleases in serum [12]. Secondly, the cellular uptake capacity of anionic CRISPR/Cas9 is rather low because of electrostatic repulsion against negatively charged cell membranes [13]. Thus, it is urgent to develop safe and efficient CRISPR/Cas9 delivery systems for use in vivo.

A perfect CRISPR/Cas9 carrier should have the following important features for efficient systemic CRISPR/Cas9 delivery. The carrier must have a sufficiently long circulation time to ensure effective tumor accumulation of the CRISPR/Cas9 system [14]. The gene carrier should also have efficient penetration into tumor tissues [15]. Moreover, the vehicle should have adequately high cellular uptake and efficient endosomal escape to allow CRISPR/Cas9 system to be released from the endosome, resulting in efficient cytoplasm delivery [16]. Most important of all, the CRISPR/Cas9 carrier must have low toxicity to ensure its safe application in clinical trials [17]. Recently, the appearance of nanomedicine seems to offer a potential option to overcome the challenges of using the CRISPR/Cas9 carrier. A considerable number of researches in nanoparticles (NPs) have been conducted regarding their possible applications in multipurpose diagnosis and therapy of cancer, because NPs show good biocompatibility and payload capacity [18,19,20]. More and more scientists tend to wrap CRISPR/Cas9 with NPs to achieve its safe and efficient delivery [21,22]. In this review, we will describe the composition of and delivery methods for the CRISPR/Cas9 system, and especially introduce different kinds of NPs for delivery of the CRISPR/Cas9 system, underlining the applications of NPs-based CRISPR/Cas9 systems in the treatment of tumors. Finally, the potential of CRISPR/Cas9 technology in cancer studies will be outlined.

## 2. CRISPR/Cas9 System

CRISPR/Cas9 system is the third generation of site-directed genome editing technology after zinc finger nucleases and transcription-activator effectors [23]. Clustered regularly interspaced short palindromic repeats (CRISPR) were first found in *E. coli* in 1987 [24] and later in many other bacterial species [25]. Then, it has been demonstrated that CRISPR and its CRISPR-associated protein (Cas) are related to the adaptive immunity targeting foreign viral DNA [26]. In the CRISPR/Cas9 system, invading foreign DNA is cleaved into small DNA fragments by Cas nucleases, which are then incorporated into the CRISPR locus of the host genome as spacers. In response to viral and phage infection, the spacer is used as a transcriptional template to generate crRNA. The mature crRNA is combined with transactivating crRNA (tracr-RNA) to form a tracrRNA: crRNA complex that directs Cas9 to a target site [27]. Cas9-mediated sequence-specific cleavage requires a DNA sequence protospacer matching crRNA and a short protospacer adjacent motif (PAM). For ease of application in genome editing, researchers designed a single guide RNA (sgRNA), a chimeric RNA that contains all essential crRNA and tracrRNA components [28]. Thus, the CRISPR/Cas9 system achieves target site-specific excision in the genome and can cause specific modification of endogenous sequences of genome that was formerly impossible using conventional RNAi or zinc finger nucleases, or transcription-activator effectors (Figure 1) [29,30,31]. Moreover, multiple sgRNAs can be used in CRISPR system to allow simultaneous multiple genome editing [32]. This feature, known as multiplexing, is of great value because tumors are mostly caused by many mutations in the genome. In brief, CRISPR/Cas9 technology provides a new view of oncogenes and cancer therapy as its induction is more simple and more efficient as compared to the above technologies. To date, the CRISPR/Cas9 system has been used in many in vivo and in vitro tumor models for therapeutic purposes [33].

### 2.1. Delivery of CRISPR/Cas9 System

Usually, the CRISPR/Cas9 system can be efficiently delivered into cells using physical methods or delivery vectors [34]. Physical methods, including microinjection and electroporation, are more suitable for in vitro CRISPR/Cas9 based genome editing therapy. Delivery vectors can be mainly classified as viral and non-viral vectors. Although viral vectors have realized efficient delivery of the CRISPR/Cas9 cargo via cell infection, they may cause needless immunogenicity and mutations in the host, thus limiting their clinical use. Additionally, transfection commercial reagents have been used deliver CRISPR/Cas9 system in vitro. These reagents are not suitable for transfecting large fragments of plasmids and also have unpredictable cytotoxicity [35], so they cannot be widely used in clinical practice. On the other hand, NPs show good biocompatibility and low side effects because of their ultrasmall diameter and environmental basic elements [36]. Numerous studies and clinical trials have displayed that NPs can be safely used in vitro and in vivo up to now [37,38]. NPs can be taken up by cells through endocytosis to deliver CRISPR/Cas9 cargo, so the use of targeted NPs with multifunctional modification that are appropriate for cancer treatment have attracted the attention of researchers [39]. These NPs can be coupled depending on their different charge properties, which means that they can be functionalized easily by adding diversified biomolecules such as drugs, targeting ligands, protein, genes and more [40,41,42]. These diversified biomolecules contribute to improve biocompatibility and targeting of NPs in vivo. Moreover, they play important roles in combined therapies of tumors, such as combined gene and photothermal therapy [43].

Different types of NPs achieve their cellular internalization through different pathways. Most receptor-bound NPs can enter cells through receptor dependent clathrin-mediated or caveolin-mediated endocytosis, while non-targeted NPs can also undergo non-specific internalization via pinocytosis or even phagocytosis [44]. After cellular internalization, NPs are initially located within endosomes (pH 6.5–6.8), and these endosomes subsequently develop into late endosomes (pH 5.2–6.2) [45]. These late endosomes would fuse with lysosomes (pH 4.5–5.2), destroying phagocytosed material with the assistance of hydrolases and an acidic environment, so escape of NPs from endosomes is essential before lysosomal digestion [46]. Physicochemical properties of NPs such as size, structure, surface charge, and polarity directly affect the transfection efficiency, cellular uptake mechanism, and cytotoxicity [47]. Importantly, some cationic NPs incorporating polyamidoamine (PAMAM), polyethyleneimine (PEI), or chitosan, have been proposed to induce endosomal escape by a proton sponge mechanism [48,49]. When the pH in a lysosome drops, the cationic polymer in the lysosome can capture a large number of protons and cause the influx of chloride ions. Then, many water molecules are absorbed into the lysosome, resulting in the imbalance of osmotic pressure of the lysosome. Finally, lysosomes break down to deliver the endocytosis CRISPR/Cas9 system into the nucleus for gene editing (Figure 2) [50].

### 2.2. Design of NPs-Based CRISPR/Cas9 System

As we all know, Cas9 nucleases and sgRNAs are two key components for the functional activity of the CRISPR/Cas9 system. They can be prepared in different forms, such as plasmids [51], mRNAs [52], a Cas9/sgRNA complex (usually called ribonucleoprotein (RNP)) [53,54], etc.

When the CRISPR/Cas9 system is produced in the form of plasmids, either a plasmid that encodes for both Cas9 and sgRNA or two separate plasmids each encoding Cas9 or sgRNA can be selected [55]. A plasmid encoding for all of the CRISPR/Cas9 elements enters the nucleus to transcribe into mRNA, and then enters the cytoplasm to translate into Cas9 protein for gene editing [56]. It is an attractive strategy due to its simplicity and low cost. However, a plasmid encoding for all of the CRISPR/Cas9 elements will have a relatively large size [57], which is a challenge for the load capacity of NPs, causing difficulties in the modification of NPs and delivery in vivo. In addition, a plasmid translating all elements required for CRISPR/Cas9 system requires an extended period, which will cause potentially generating off-target insertions or deletions [58]. As an example, Sakuma et al. constructed seven gRNA expression cassettes and a Cas expression cassette in the same plasmid which was transcribed and translated in cells to form the corresponding seven gRNAs and a Cas9 protein to realize multi-gene editing of a single target, and the editing efficiency was between 4% and 36% [59]. Two separate plasmids each encoding Cas9 or sgRNA avoid the large size of plasmid, but their off-target effects are still not properly resolved.

A CRISPR/Cas9 system in the form of mRNA (Cas9 mRNA and sgRNA) can bypass DNA transcription to improve its gene editing efficiency [60]. Moreover, the expression of Cas9 mRNA is transient in the nucleus, and eventually Cas9 mRNA can be completely removed from the nucleus, which effectively avoids mutations to similar sites in the genome caused by Cas9 mRNA long-term retention in the nucleus and is beneficial to reduce off-target effects [61]. Liang et al. compared the editing efficiency of Cas expressing plasmid, Cas mRNA and Cas RNP in a variety of cells, and found that Cas mRNA transfection produced the highest gene editing efficiency in most cell lines [58]. Nevertheless, the poor stability of Cas9 mRNA in vivo causes its inability to exert continuous effects and limits its wide applications in gene editing.

Since the final objective is to deliver the Cas9/sgRNA complex to the nucleus of the host cell, RNP that needs the least amount of intracellular processing may be considered as the most straightforward strategy [54]. The sgRNA is easily destroyed in vivo [62], and RNP assembly can provide significant protection for the sgRNA [63]. Most important of all, RNP also can provide transient and efficient gene editing, with relatively less off-target indels being generated [64,65]. However, RNP needs to be ready-to-use and cannot be stored for a long time, which places great demands on the protective ability of its carrier. Therefore, investigators must give enough thought to the advantages and disadvantages of each system when making designing NPs-based CRISPR/Cas9 systems.

## 3. Kinds of NPs-Based CRISPR/Cas9 System

Various nanosystems are constantly being developed for efficient delivery of the CRISPR/Cas9 system both in vitro and in vivo [12,66,67]. It is believed that the biological and clinical applications of the CRISPR/Cas9 system will be more extensive and safer with the continuous optimization of nano-delivery systems. According to the classification of nanomaterials, the NPs for systemic CRISPR/Cas9 delivery can be divided into lipid NPs (LNPs), polymer NPs (PNPs), inorganic NPs (INPs) and NPs of other structures (Figure 3 and Table 1). 

### 3.1. LNPs

Lipids are a diverse group of naturally occurring or synthetic organic compounds which include fatty acids and their derivatives [68]. Amphiphilic phospholipids are a major component of cell membranes and liposomes [69]. Therefore, liposomes as three-dimensional spherical structures that encapsulate hydrophilic or hydrophobic molecules can enter cells by endocytosis [70]. When a liposome is positively charged, DNA and RNA can form strong bonds through electrostatic attraction, adsorbing them on the surface of liposome [71], which efficiently protects the nucleic acid from degradation. Currently, commercial cationic liposomes and lipofection kits are available for the delivery of RNAi gene and plasmid DNA (pDNA). This means that LNPs with good biocompatibility can also apply to the delivery of the CRISPR/Cas9 system, in which the gene editing efficiency reached 80% in human cells [66]. 

Targeting of LNPs first requires the process of transportation in vivo. Targeting of LNPs is reflected in two aspects: the effective enrichment in tumor tissue and the efficient penetration into tumor cells. Simple targeted modifications and good membrane fusion ability of LNPs can help to concentrate the CRISPR/Cas9 system in tumor tissue and enable efficient penetration of cell membranes, thereby significantly improving the gene silencing efficiency of the CRISPR/Cas9 system [72]. Encouragingly, researchers have conducted extensive research to overcome this problem. Li et al. designed cationic liposomes modified with R8-DGR, whose fluorescence at different depths was significantly higher than other liposomes in the 3D tumor spheroid model, showing good tumor deep penetration ability and contributing to tumor eradication by targeted delivery of CRISPR/Cas9 system to knockdown of hypoxia-inducible factor-1 alpha [73]. Rosenblum et al. used a novel amino-ionizable lipid nanoparticle for the delivery of Cas9 mRNA and sgRNAs. Epidermal growth factor receptor (EGFR) antibody was conjugated to this lipid nanoparticle to improve its targeting in disseminated tumors [74]. Cheng et al. reported that selective targeted function of LNPs could be achieved by adding a complementary selective organ targeting (SORT) molecule. Results showed that the percentage of 1,2-dioleoyl-3-trimethylammonium-propane (DOTAP) was the key factor that tuned tissue specificity. LNPs moved progressively from liver to spleen, and then to lung with increasing molar percentage of DOTAP, demonstrating a clear and precise organ-specific delivery trend with a threshold that allowed exclusive lung delivery. This methodology could enable specific CRISPR/Cas9 gene editing of not only liver, lung, and spleen but also various specific types of cells within them, such as epithelial cells, endothelial cells, B cells and T cells. Additionally, this strategy was compatible with multiple forms of CRISPR/Cas9 system, including mRNA, Cas9 mRNA/single guide RNA and Cas9 ribonucleoprotein complexes [75]. Liu et al. took inspiration from cell membranes consisting of natural phospholipids, and developed multi-tailed ionizable phospholipids (iPhos) for delivering messenger RNA or mRNA/single-guide RNA in vivo. Interestingly, the chemical structure of iPhos could control organ selectivity in vivo, where iPhos lipids synergistically function with different kinds of helper lipids to prepare multicomponent LNPs for tissue-selective mRNA delivery and CRISPR/Cas9 gene editing in spleen, liver and lungs, respectively [76].

In recent years, tumor microenvironment (TME) responsive liposomes have received attention from many researchers [77]. These liposomes underwent a conformational change when exposed to microenvironmental stimuli (such as pH, reactive oxygen species (ROS), temperature, magnetic field or enzyme) in order to achieve specific release of the CRISPR/Cas9 system and improve gene editing efficiency [78]. For example, Zhen et al. constructed pH-sensitive cationic liposomes that produced specific responses to normal tissues and tumor sites with different pH values, helping efficient CRISPR/Cas9 system release in tumor sites [79]. Yin et al. developed an ultrasound-controlled CRISPR/Cas9 release system, in which the hematoporphyrin monomethyl ether (HMME) yielded abundant ROS to damage tumor cells under ultrasound irradiation, and meanwhile the generated ROS could induce lysosomal rupture to release RNP and destroy the oxidative stress-defense system, ultimately significantly promoting tumor cell apoptosis [80].

### 3.2. PNPs 

Cationic polymers can effectively compress negatively charged nucleic acid through electrostatic interaction to obtain PNPs, which can significantly protect sgRNA and Cas proteins from degradation, improve delivery targeting, and control drug release rate in vivo [81]. At present, the most widely used cationic polymers are PEI and chitosan [82]. Zhang et al. constructed PEI-β-cyclodextrin cationic polymers to deliver the CRISPR/Cas9 system, and these cationic polymers greatly improved editing efficiency via the electrostatic compression of negatively charged pDNA [81]. Jo et al. designed poly lactic-co-glycolic acid (PLGA) NPs to deliver a model CRISPR/Cas9 plasmid into primary bone marrow derived macrophages. The diameter of engineered PLGA NPs was approximately 160 nm, which facilitated efficient transfer in vitro [83].

Meanwhile, cationic PNPs also have a proton sponge effect, which can realize lysosome escape and release endocytosis genes into cytoplasm, and then achieve markable gene editing efficiency [84]. Liu et al. designed a CRISPR/Cas9 system delivered by a cationic polymer made of phenylboronic acid modified PEI. It was shown that the co-localization of pDNA and lysosome was reduced after incubation for 4 h, which proved that these PNPs could deliver the CRISPR/Cas9 system to cytoplasm to achieve remarkable therapeutic effects on tumors via a proton sponge effect [12].

Targeted modification of PNPs can also observably increase drug accumulation in tumor tissues and uptake by tumor cells to reduce side effects [85,86]. Zhang et al. developed lactose acid-modified chitosan NPs to deliver the CRISPR/Cas9 system for the knockout of the vascular endothelial growth factor 2 (VEGFR2) gene in liver tumor cells, and they demonstrated that the average intratumoral fluorescence intensity of these NPs at different time points was stronger than that of free labeled dyes, which indicated that this nanosystem carried out effective drug delivery and enhanced tumor accumulation for great curative effects of hepatocellular carcinoma (HCC) [85]. Nguyen et al. reported that stabilized Cas9 RNPs in polyglutamic acid-modified NPs could improve editing efficiency by approximately twofold, reduce toxicity, and enable lyophilized storage without loss of activity [86].

### 3.3. INPs

In recent years, INPs such as mesoporous silica NPs and metal NPs, have shown the potential to be applied as CRISPR/Cas9 delivery systems as well, due to their good physiological compatibility, optical characteristics and large specific surface area [87]. Zhang et al. prepared NPs with mesoporous silica as the shell, and found that these NPs displayed great biocompatibility in vivo and editing efficiency of EGFR of more than 60%, further achieving effective tumor gene therapy [88]. Gold NPs have unique photothermal and structural properties [89]. Their surface plasmon resonance can generate heat to trigger the release of the CRISPR/Cas9 system. Thus, Wang et al. designed photothermal triggered lipid-encapsulated gold NPs for systematic delivery of CRISPR/Cas9 [67]. Magnetic NPs under the controlled magnetic field are expected to greatly accelerate the delivery of CRISPR/Cas9 system into cells, further improving gene editing efficiency. Rohiwal et al. synthesized PEI-coated magnetic iron oxide NPs as a delivery system for plasmids encoding CRISPR/Cas9, and they demonstrated that these magnetic NPs could significantly improve the safety and utility of gene editing [90].

### 3.4. NPs of Other Structures

Nanogel is a nanopolymer gel composed of hydrophilic or amphiphilic polymer chains. Its internal crosslinking three-dimensional network of nanostructures with high structure stability can avoid the CRISPR/Cas9 system disintegrating or gathering and further reducing CRISPR/Cas9 system release in the blood due to the dilution and interaction with various components in the blood flow [91]. Chen et al. designed novel core-shell nanostructure, liposome-templated hydrogel NPs (LHNPs) for efficient codelivery of Cas9 protein and nucleic acids. The results suggested that LHNPs could release 91.5% sgRNA and 85.2% Cas protein within 3 days, showing good controlled release ability [92].

Nanowire is yarn-like NPs synthesized by rolling cycle amplification [93]. Base pairing between nanowires and Cas/sgRNA complexes through complementary sequences produces strong but reversible interactions that maintain cell viability while achieving target gene destruction, thereby improving gene editing efficiency [94]. Sun et al. developed DNA nanowires to balance the binding and release of Cas/sgRNA complex. Experiments showed that the gene editing efficiency of PEI-coated common carrier was only 5%, while the gene editing efficiency of the host constructed by DNA nanowires was 36%, which provided a design idea for improving the therapeutic effect of tumors [94].

The unique electronic structure and minimal atomic thickness of nanosheets, with the resulting large surface-area-to-thickness ratio, have attracted great attention in a range of areas [95], including in the delivery of the CRISPR/Cas9 system. The two-dimensional folded honeycomb structure of the emerging material black phosphorus provides a large specific surface area, and the periodic atomic grooves on the surface of black phosphorus provide an ideal anchor site for Cas9 protein, exhibiting the potential of loading and delivery of biomolecules. Zhou et al. constructed black phosphorus nanosheets with loading capacity of Cas ribosomal protein reaching 98.7%, which can achieve effective tumor-related gene editing at a lower concentration of NPs, reflecting good loading capacity [96].

## 4. Applications of NPs-Based CRISPR/Cas9 System in Cancer Therapy

The most common cancer treatments include chemotherapy, radiotherapy and surgery. However, these therapies all have severe side effects and bring a highly unpleasant treatment experience to patients. Thus, several novel therapies, such as gene therapy [97], immunotherapy [98] and photothermal therapy [99], have emerged and also achieve amazing curative effects. The genome altering capacities of CRISPR/Cas9 technology demonstrate its utility in cancer therapy, and it can accelerate the development of tumor therapeutics in combination with the research and development into nanotechnology. The NPs-based CRISPR/Cas9 system is able to be used in various treatments, and is especially suitable for gene therapy [33]. Below, we will introduce the applications of NPs-based CRISPR/Cas9 system in cancer therapy in detail. 

### 4.1. Gene Therapy

Genes encoding growth factors and their receptors, transcription factors, signal transducers, and chromatin remodeling proteins have oncogenic potential. CRISPR/Cas9 can directly target these oncogenes, knocking them out and inhibiting cancer growth through different mechanisms. For instances, polo-like kinase 1 (PLK1) is a conserved mitotic serine-threonine protein kinase, which functions as a regulatory protein, and is involved in the progression of the mitotic cycle. It plays important roles in the regulation of cell division, maintenance of genome stability, in spindle assembly, mitosis, and DNA-damage response [100]. Rosenblum et al. reported that a single intracerebral injection of CRISPR-LNPs against PLK1 into aggressive orthotopic glioblastoma enabled up to approximately 70% gene editing in vivo, which caused tumor cell apoptosis, inhibited tumor growth by 50%, and improved survival by 30%. To reach disseminated tumors, these CRISPR-LNPs were also engineered for antibody-targeted delivery. Intraperitoneal injections of EGFR-targeted sgPLK1-cLNPs caused their selective uptake into disseminated ovarian tumors, enabled up to approximately 80% gene editing in vivo, inhibited tumor growth and increased survival by 80% [74]. Similarly, Chen et al. developed LHNPs for efficient co-delivery of CRISPR/Cas9 targeting PLK1, and they found that these LHNPs effectively inhibited tumor growth and improved tumor-bearing mouse survival [92].

Tumor suppressor genes, also known as anti-cancer genes, are a class of genes that exist in normal cells and can inhibit cell growth and have potential tumor suppressor effects. Tumor suppressor genes play a very important negative regulatory role in the control of cell growth, proliferation and differentiation. They interact with oncogenes to maintain the relative stability of positive and negative regulatory signals. When these genes are mutated, deleted or inactivated, they can cause malignant transformation of cells and lead to the occurrence of tumors. The most common tumor suppressor genes are Rb, P53, APC and other genes [101]. Ju et al. designed the self-assembly of gold NPs with SpCas9 protein (SpCas9-AuNCs) for efficient knockout of the E6 oncogene, thus restoring the function of P53 and inducing apoptosis in cervical cancer cells with little effect in normal human cells [102].

### 4.2. Chemotherapy

Chemotherapy is still the most common treatment for tumors. Several studies have suggested that genome editing using CRISPR/Cas9 is helpful for enhancing efficacy of chemotherapy [103]. EGFR and VEGFR2 play the important role in various tumors, leading to the growth and proliferation of tumor cells. Zhang et al. reported on a lactobionic acid functionalized and stimuli-responsive chitosan based nanocomplex to co-deliver sgVEGFR2/Cas9 plasmid and paclitaxel for combined treatment of HCC. The genome editing efficiency of sgVEGFR2/Cas9 in the nanosystem achieved up to 38.6% of HepG2 cells in vitro and 33.4% of tumor tissues in vivo. The nanocomplex suppressed more than 60% VEGFR2 protein expression of HepG2 cells and inhibited HCC progress by 70% in mice. In vivo study indicated the obvious tumor accumulation and the good biosafety of this nanosystem [85]. Moreover, they developed hollow mesoporous silica NPs (HMSNs) to carry Sorafenib and CRISPR/Cas9 system targeting EGFR for combined therapy of HCC, and proved it was also a promising approach to enhance HCC treatment efficiency [88]. Epirubicin as a topoisomerase inhibitor is a potential anthracycline for the treatment of head and neck cancer (HNC) [104]. Human antigen R (HuR) is an RNA binding protein encoded by the ELAVL1 gene, which plays an important role in facilitating tumor survival, invasion and resistance [105]. Thus, Wang et al. designed multifunctional NPs modified with pH-sensitive epidermal EGFR-targeting and nuclear-directed polypeptides for efficient delivery of HuR CRISPR/Cas9 and epirubicin to SAS cell line and SAS tumor-bearing mice. Results showed that the cellular uptake and transfection efficiency of these NPs in vitro was remarkably enhanced by EGFR targeting, ligand-mediated endocytosis, and endosomal escape. Meanwhile, knockout of HuR using CRISPR/Cas9 NPs significantly inhibited tumor growth and improved the survival percentage in epirubicin-treated SAS tumor-bearing mice [106].

Drug resistance of tumor cells is always a headache problem in clinical chemotherapy. Li et al. developed a CRISPR/Cas9 nanoeditor to knockout two key oncogenes E6 and E7 in order to alleviate chemotherapy-resistance in cervical cancer. Results showed that these cationic LNPs combined with chemotherapy drug docetaxel could significantly inhibit the drug tolerance of cancer cells and improve the therapeutic effect on cervical cancer [107].

### 4.3. Immunotherapy

Programmed cell death protein-1 (PD-1), an immune checkpoint regulator, is a T-cell receptor whose role is to inhibit T-cell activation, thereby regulating immune tolerance and reducing autoimmune responses, while also allowing cancer cell immune escape [108]. Antibodies that neutralize PD-1 or its ligand (PD-L1) have been successfully used in the treatment of several cancers, especially advanced non-small-cell lung cancer [109,110]. To develop an alternative treatment method based on immune checkpoint blockade, Deng et al. designed a novel and efficient CRISPR/Cas9 genome editing system delivered by cationic copolymer aPBAE to downregulate PD-L1 expression on tumor cells via specifically knocking out Cyclin-dependent kinase 5 (Cdk5) gene in vivo. The expression of PD-L1 on tumor cells was significantly attenuated by knocking out Cdk5, leading to effective tumor growth inhibition in murine melanoma and lung metastasis suppression in triple-negative breast cancer. Importantly, we demonstrated that aPBAE/Cas9-Cdk5 treatment elicited strong T cell-mediated immune responses in tumor microenvironment and that the population of CD8^+^ T cells was significantly increased while regulatory T cells (Tregs) was decreased. This may be the first case to exhibit direct in vivo PD-L1 downregulation via CRISPR-Cas9 genome editing technology for cancer therapy. It will provide a promising strategy for preclinical antitumor treatment through the combination of nanotechnology and genome engineering [111]. Zhou et al. used flow cytometry, expression analyses and co-culture systems to explore hepatocellular CCRK/EZH2/NF-κB/IL-6 signaling that mitigated anti-tumor T cell responses by induction of MDSC immunosuppression. Then, they employed lipid/calcium/phosphate NPs for the delivery of plasmid DNA encoding an IL-6 protein in the CCRK knockout orthotopic HCC model using CRISPR/Cas9 technology, and found that targeted inhibition of CCRK enhanced the efficacy of anti-PD-L1 in HCC via abrogation of MDSC immunosuppression [112].

### 4.4. Other Therapy

The NPs-based CRISPR/Cas9 system also displays its special capacities in other cancer treatments. Wang et al. developed gold NPs carrying CRISPR/Cas9 targeting PLK-1 for photothermal therapy owing to the unique photothermal and structural properties of gold NPs. After laser irradiation for a period of time, the CRISPR/Cas9 system released by the photothermal effect could down-regulate the PLK-1 protein by 65%, further achieving greater therapeutic effects [67]. Li et al. reported a proton-activatable DNA-based nanosystem for co-delivery of PLK1 Cas9/sgRNA and DNA zyme in the combined therapy of breast cancer. The sgRNA recognition sequence, DNA zyme sequence and HhaI enzyme cleavage site in Cas9/sgRNA were integrated in an ultra-long ssDNA strand as the scaffold of the nanosystem. The DNA strands were compressed with the DNA zyme cofactor Mn^2+^ to form NPs, and the HhaI enzyme coated with an acid-degradable polymer was assembled on the surface of the NPs. Under the action of protons in the lysosome, the polymer envelope is decomposed, and the HhaI enzyme is exposed, which recognizes and cuts off the cleavage site; this triggers the release of Cas9/sgRNA and DNA zyme to regulate gene expression, so as to obtain a tumor suppression rate of about 85% in the tumor xenografts model via inducing the apoptosis of breast cancer cells [113]. Yin et al. designed the HMME@Lip-Cas9 to knock down NFE2L2, thereby alleviating the adverse effects and augmenting the therapeutic efficiency of sonodynamic therapy (SDT), which provided a synergistic therapeutic modality in the combination of SDT with gene editing for HCC [80].

## 5. Potentials of CRISPR/Cas9 in Oncology

CRISPR/Cas9 technology has huge application potentials in biology. Therefore, we focus on introducing some research contributions of CRISPR/Cas9 in oncology, mainly including clinical trials, CAR-T therapy and tumor organology in this review. 

### 5.1. Clinical Trials and CAR-T Therapy

The first clinical trial using the CRISPR/Cas9 system for cancer therapy recruited its first patient at Sichuan University’s West society China Hospital in Chengdu in 2016 [114]. In this non-randomized, open-label phase I study (NCT02793856), the safety properties of PD-1 knockout engineered T cells in vitro in metastatic non-small cell lung cancer that has progressed after all standard therapy were assessed. Patients participating in the gene editing trial provided peripheral blood lymphocytes. The CRISPR/Cas9 system was used to perform in vitro experiments on PD-1 knockout T cells. The edited lymphocytes are screened, expanded, and then infused back into the patient (Figure 4). 

To date, there have been 17 clinical trials of the CRISPR/Cas system as an oncology therapeutic intervention (Table 2). In a phase I clinical study (NCT03164135), a novel intracellular immune checkpoint (CISH) was programmed to inhibit its expression, resulting in a powerful cytokine immune response to achieve currently unachievable inhibition of intracellular checkpoint.

The vast majority of the trials have been applied to the chimeric antigen receptor (CAR)-T cell system by delivering precise and effective gene editing of human T cells to humans without sacrificing cell viability or function [115]. Recent advances in cancer immunotherapy have revealed the importance of CAR-T cells [116,117,118]. CAR-T cell therapy targeting CD19, a cell surface molecule, has shown positive responses in patients with relapsed B-cell malignancies [119,120]. In a recent preclinical study, the tumor rejection activity was significantly enhanced using CRISPR/Cas9 technology to deliver a CAR-targeted gene to the T-cell receptor alpha-chain site compared with using CAR-T cells randomly produced integrated vectors [121]. However, the CAR-T cell therapy for solid tumor is not mature enough. This treatment also has some side effects, such as neurotoxicity [122], B-cell aplasia [123], and cytokine release syndrome [124]. To alleviate these side effects of CAR-T cell therapy, the CRISPR/Cas9 technique was used to remove some genes of cell death proteins or T-cell inhibitory receptors to modify primary human T cells [125]. For an example, cytotoxic T lymphocyte-associated protein (CTLA-4) plays a role in the production of T-cell inhibitory receptors. The knockout of CTLA-4 in human T-cells was realized using the CRISPR/Cas9 system [126]; T-cells deficient in both PD-1 and CTLA-4 enabled T-cell production and increased T-cell efficiency in CAR-T cell therapy at the same time [127].

Compared with in vitro genome editing, in vivo genome editing is still less used in clinical trials. It is believed that improved specificity of Cas9 targeting defined genomic regions without off-target effects, as well as the advancement of CRISPR/Cas9 delivery by way of nanosystems targeting specific organs or tumors, may certainly drive the development of further in vivo gene editing clinical trials for cancers in the future.

### 5.2. Generation of Organoid Cancer Models

Organoids derived from adult stem cells are an increasingly popular in vitro model of intact and diseased human epithelia [128], which help researchers to better understand the mechanism of tumorigenesis. Stem cells from a variety of adult tissue types can be isolated and cultured in 3D, and then proliferate, differentiate and form organoids in cell culture dishes via stimulation with tissue-specific growth factors. This provides new possibilities for research into tumor development and progression in vitro [129].

Schwank et al. pioneered the use of the CRISPR/Cas9 system in mouse-derived intestinal organoids [130]. Subsequent studies reported the transformation of healthy human colon organoids into cancer organoids by using CRISPR/Cas9 system to reproduce the adenoma-cancer sequence of colon cancer [131,132]. Knockout of tumor suppressors such as APC, TP53 and SMAD4 as well as gene editing of oncogenes such as KRAS and PI3K using the CRISPR/Cas9 system facilitated the establishment of colon cancer organoid models in vitro. Additionally, Drost et al. also established a mismatch repair-deficient colorectal cancer model by deleting MLH1 DNA repair genes in colonic organoids [133]. The above results revealed great potential for the use of the CRISPR/Cas9 system in oncology research, especially since the successful establishment of the organoid cancer models could help researchers to explore the treatment of tumors from a new perspective.

## 6. Future Perspective and Challenges

In 2013, the CRISPR/Cas9 system was first introduced into mammalian cells for genome editing, which is expected to bring revolutionary changes to the medical field [8,9]. However, there are still some problems in the application of the CRISPR/Cas9 system in clinical work. Above all, the sequence of CRISPR/Cas9 is relatively long, resulting in reduced gene editing efficiency. Secondly, because the spatial conformation of chromosomes can reduce the accessibility of some target genes, gene editing may be difficult to achieve. Thirdly, the unexpected expression of the CRISPR/Cas9 system in non-target tissues and organs will reduce the safety of gene editing. Thus, it is imperative to find safe and efficient medicinal materials for the delivery of the CRISPR/Cas9 system. To date, the use of commercial liposomes to transfect the CRISPR/Cas9 system into cells has been very mature. The nano-delivery system as the most potential vector of the CRISPR/Cas9 system shows lower cytotoxicity compared with commercial liposome. However, the application of the CRISPR/Cas9 system in vivo is unsatisfactory. Although only a few NPs-based CRISPR/Cas9 systems have been tested intravenously [134,135], more formulations that have not yet been tested are being considered for this potential application. On the one hand, LNPs have the lead in terms of safety of CRISPR/Cas9 system delivery in vivo owing to their great biocompatibility. Especially, the development of TME responsive LNPs is the trend with the increasing awareness of the importance of the TME in tumor therapy. On the other hand, PNPs have the advantage in targeting of the CRISPR/Cas9 system delivery in vivo because their special physicochemical properties make them easier to be modified by various biomolecules. Moreover, INPs have an irreplaceable position in the combination therapy of tumors. For example, when combining gene therapy using the CRISPR/Cas9 system with magnetic hyperthermia, magnetic INPs should be selected as the vector of the CRISPR/Cas9 system [136]. NPs of other structures also play important roles in the efficient delivery of CRISPR/Cas9 system because their complex nano-structure can protect it. These unique properties of different types of NPs have great value in safely and efficiently delivering CRISPR/Cas9 both in vitro and in vivo. It is a major challenge for researchers to make full use of these properties to prepare suitable carriers for CRISPR/Cas9. Meanwhile, we believe that the multifunctional nano-delivery system that combines different properties of multiple NPs will be a better choice for the delivery of CRISPR/Cas9 in the future. For an example, Wang et al. not only used the excellent photothermal effect of gold NPs to deliver CRISPR/Cas9 in photothermal therapy, but also used the good biocompatibility of liposomes to increase cellular uptake and improve safety via wrapping the photothermal triggered gold NPs under the lipid layer [67]. With the rapid development of these multifunctional nano-delivery systems, it is believed that the CRISPR/Cas9 system will be widely used in the biomedical field in the future.

The potentials of CRISPR/Cas9 technology in oncology have yet to be fully tapped. Currently, CAR-T therapy has a major limitation, that is it is more suitable for the treatment of hematological tumors than that of solid tumors [117]. The most important reason is that T cells cannot effectively reach the tumor tissues in vivo [116]. There is no literature report on the application of NPs to deliver CRISPR/Cas9 system to transform T cells, but it is trusted that the combined use of nanotechnology and the CRISPR/Cas9 system is the most promising way to overcome this flaw of CAR-T therapy. Combining CRISPR/Cas9 technology with advanced organoid cultures will enable cancer researchers to replicate the oncogene sequences of many cancer entities. This will provide deeper insights into biological effects caused by individual mutations and separate driver mutations from passenger mutations. In addition, we expect these clinical trials of CRISPR/Cas9 will achieve desirable outcomes. In a word, to further improve the in vivo application of the CRISPR/Cas9 system, it is vital to improve the associated nano-delivery system to lay the foundation for future therapeutic applications. In conclusion, we believe that the joint development of CRISPR/Cas9 technology and nanotechnology has contributed to and will greatly accelerate cancer research and therapy in many fields.

## Figures and Tables

**Figure 1 biomolecules-12-01239-f001:**
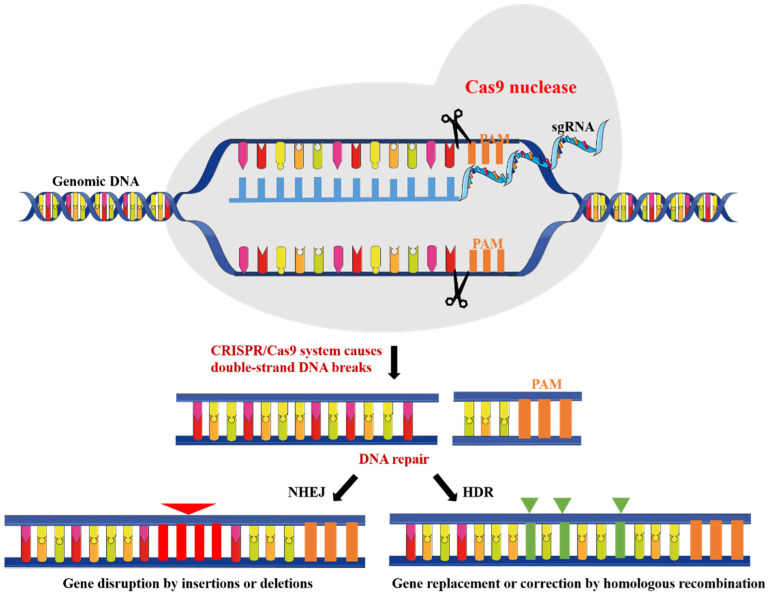
The composition and function of CRISPR/Cas9 system: the Cas9 nuclease is directed to the target DNA by complementary base-pairing with its bound sgRNA. The target site must be followed by a 3′ PAM sequence. The following cleavage of double-strand DNA triggers either the error prone non-homologous end joining (NHEJ) or homology directed repair (HDR) mechanisms.

**Figure 2 biomolecules-12-01239-f002:**
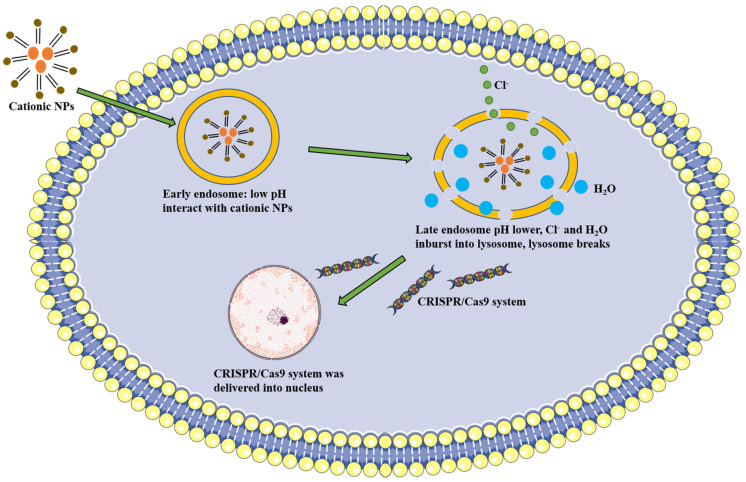
Endosomal escape of cationic NPs by the proton sponge mechanism: the cationic NPs enter early lysosomes through endocytosis, then the drop of pH coupled with the influx of chloride ions and water molecules cause the rupture of late lysosomes, thereby delivering the CRISPR/Cas9 system into the nucleus.

**Figure 3 biomolecules-12-01239-f003:**
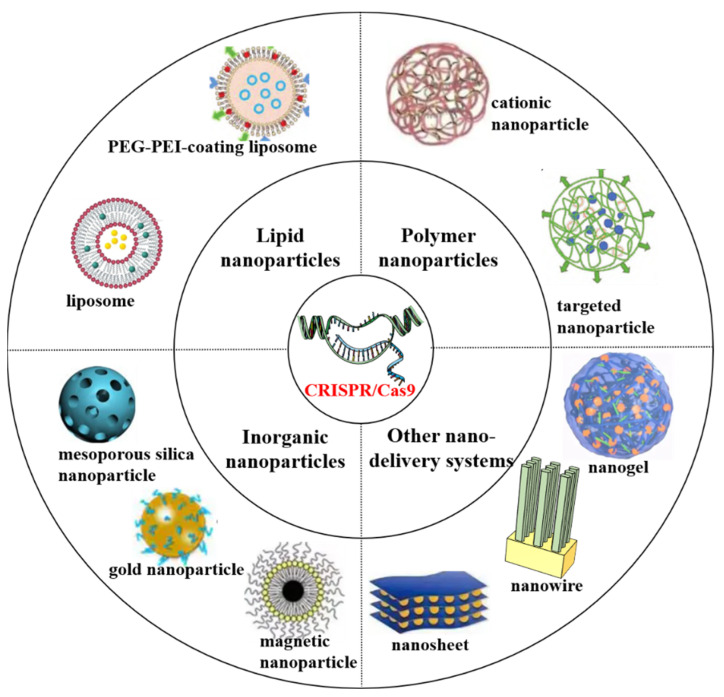
Different types of NPs-based CRISPR/Cas9 system.

**Figure 4 biomolecules-12-01239-f004:**
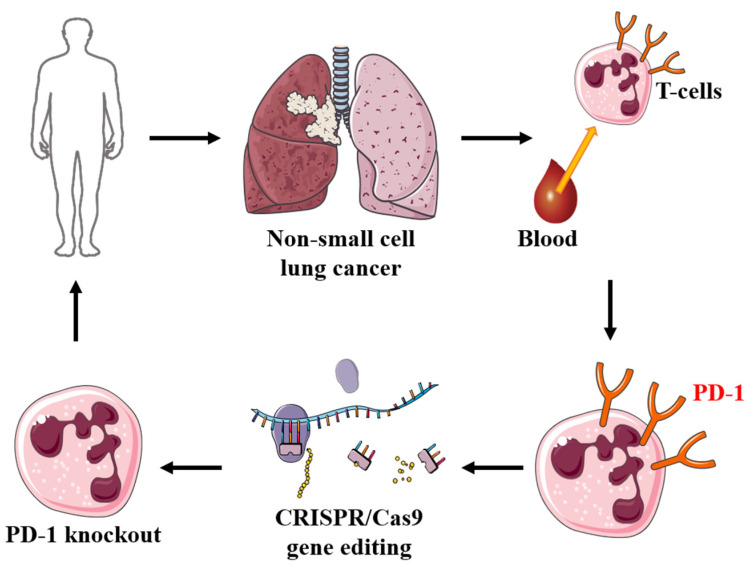
Targeted *PD-1* gene modification in T cells in the first clinical trial of non-small cell lung cancer using CRISPR/Cas9 technology: peripheral blood lymphocytes were collected from the patient with the non-small cell lung cancer. The CRISPR/Cas9 mediated knockout of the immune checkpoint gene *PD-1* was performed in human T-cells. The PD-1 knockout T-cells were expanded in vitro and then transfused back to the patient, thereby inducing an immunological response against tumor cells.

**Table 1 biomolecules-12-01239-t001:** NPs-based CRISPR/Cas9 system.

Delivery Approaches	NPs Formulation	CRISPR/Cas9 Cargo	Efficiency	Application	Reference
LNPs	Cationic liposomes	Cas9 sgRNA complexes	80%	in vitro and in vivo	[66]
	Cationic liposomes modified with R8-DGR	sgRNA	--	in vitro and in vivo	[73]
	Amino-ionizable lipid NPs	Cas9 mRNA and sgRNA	~70% (aggressive orthotopic glioblastoma) ~80% (disseminated ovarian tumors)	in vitro and in vivo	[74]
	SORT	Cas9 mRNA and sgRNA	40% (epithelial cells) 65% (endothelial cells) 12% (B cells) 10% (T cells) 93% (hepatocytes) 20% (liver) 50% (lung) 30% (spleen)	in vitro and in vivo	[75]
	iPhos	Cas9 mRNA and sgRNA	~91% (hepatocytes) ~34% (lung endothelial cells) ~20% (lung epithelial cells) ~13% (lung immune cells) ~30% (splenic macrophages) 6% (Splenic B cells)	in vivo	[76]
	pH-sensitive cationic liposomes	Cas9 mRNA and sgRNA	--	in vitro and in vivo	[79]
	HMME@Lip-Cas9	RNP	17.28% (HMME@Lip-Cas9) 58.77% (HMME@Lip-Cas9 + ultrasound)	in vitro and in vivo	[80]
PNPs	PEI-β-cyclodextrin cationic polymers	pDNA	19.1% (HBB locus) 7% (RHBDF1 locus)	in vitro	[81]
	Poly lactic-co-glycolic acid (PLGA) NPs	pDNA	95% (murine bone marrow derived macrophages)	in vitro	[83]
	MDNP	pDNA	--	in vitro and in vivo	[12]
	Polyglutamic acid-modified NPs	Cas9 RNPs	~2 fold increase	in vitro	[86]
INPs	SEHPA NPs	RNP	>60% (EGFR editing efficiency)	in vitro and in vivo	[88]
	LACP	RNP	68%	in vitro and in vivo	[67]
	PEI-coated magnetic Fe_3_O_4_ NPs	pDNA	13% (with magnetic field) 10% (without magnetic field)	in vitro	[90]
NPs of other structures	LHNPs	Cas9 protein and minicircle gRNA	1.3 times more efficiently than Lip2k	in vitro and in vivo	[92]
	DNA nanowires	Cas9 protein and sgRNA	36%	in vitro and in vivo	[94]
	Cas9 N3BPs	Cas9 sgRNA complexes	26.7% (Target 1) 32.1% (GRIN2B)	in vitro and in vivo	[96]

**Table 2 biomolecules-12-01239-t002:** Clinical trials of the CRISPR/Cas system for cancer therapy.

Identifier	Target Gene	Phase	Condition
NCT03057912	*HPV E6/E7*	I	Human Papillomavirus-Related Malignant Neoplasm
NCT03164135	*CISH*	I/II	Gastrointestinal Epithelial Cancer, Gastrointestinal Neoplasms
NCT04976218	*TGF-β*	I	Solid Tumor
NCT04767308	*CD5*	I	Relapsed/Refractory Hematopoietic Malignancies
NCT03545815	*PD-1*, *TCR*	I	Solid Tumor, Adult
NCT05066165	*WT1*	I/II	Acute Myeloid Leukemia
NCT05309733	*CD33*		Leukemia, Myeloid, Acute
NCT03747965	*PD-1*	I	Solid Tumor, Adult
NCT04035434	*CD19*	I	B-cell Malignancy Non-Hodgkin Lymphoma B-cell Lymphoma Adult B Cell ALL
NCT04502446	*CD70*	I	T Cell Lymphoma
NCT03081715	*PD-1*	--	Esophageal Cancer
NCT05037669	*CD19*	I	Acute Lymphoblastic Leukemia, Chronic Lymphocytic Leukemia, Non Hodgkin Lymphoma
NCT04244656	*BCMA*	I	Multiple Myeloma
NCT04438083	*CD70*	I	Renal Cell Carcinoma
NCT03166878	*CD19*	I/II	B Cell Leukemia, B Cell Lymphoma
NCT03398967	*CD19/CD20/CD22*	I/II	B Cell Leukemia, B Cell Lymphoma
NCT04557436	*CD52*, *TRAC*	I	B Acute Lymphoblastic Leukemia

## Data Availability

Not applicable.

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
