# Peer review of "Research Progress on Nanoparticles-Based CRISPR/Cas9 System for Targeted Therapy of Tumors"

_biomolecules, 2022, doi:10.3390/biom12091239_

Round 1

Reviewer 1 Report

Nie et al. have written a very comprehensive review of nanoparticles as potential delivery system for CRISPR/Cas9 designed for targeted therapy of tumors.  This is indeed a thought-provoking review written by experts in the field.  Even though since its introduction in 2013, CRISPR/Cas9 has largely been used for in vitro manipulation of cells, the potential for its use as a therapeutic alternative to chemotherapy in vivo was  in the mind of investigators.  With the rapid advancement in nanotechnology and nanomedicine, the prospects for their use in the delivery of gene therapeutics such as CRISPR/Cas9 is becoming more realistic, particularly based on some of the approaches highlighted in this review.  Thus, this review will spur more investigators to develop more nanoparticles with potential for delivering CRISPR/Cas9 into tumor cells or cells of interest in vivo.

      Having said that, there were several grammatical errors in this review that will require extensive editing:

1)     Line 35, ….. CRISPR/Cas9 system is unable to be directly applicated in … should be re-phrased. 

2)     Similarly line 39 should be re-phrased.

3)     Line 50 should be changed to …In this review, we will first describe……

4)     Line 76 should be changed to ….viral and non-viral vectors.

5)     Line 279, they should delete the words Cite examples.

6)     Line 336, delete ..we cannot describe it clearly.

7)     Line 341 delete  ‘been’ from …most have been failed…

8)  

Author Response

List of Responses
Dear Editor and Reviewers:
Thank you for your comments concerning our manuscript entitled “Research progress of nanoparticles-based CRISPR/Cas9 system for targeted therapy of tumors.” (ID:1818940). Those comments are all valuable and very helpful for revising and improving our paper. We are really sorry for our shortcomings in the manuscript. We have studied comments carefully and have made correction which we hope meet with approval. Revised portion are marked in red in the paper. The main corrections in the paper and the responds to the reviewer’s comments are as flowing:

Reviewer1.

1.Line 35, ….. CRISPR/Cas9 system is unable to be directly applicated in … should be re-phrased. 

Response: Thank you for your great comments. We have rewritten this sentence in the manuscript.

2.Similarly line 39 should be re-phrased.

Response: Your comments are very helpful for us. We have rewritten this sentence in the manuscript.

3.Line 50 should be changed to …In this review, we will first describe……

Response: Thank you for your careful check. We have completed the modification as you suggested.

4.Line 76 should be changed to ….viral and non-viral vectors.

Response: Thank you for your careful work. We have completed the modification in the manuscript.

  1. Line 279, they should delete the words Cite examples.

Response: We have completed the modification for your suggestion.

6.Line 336, delete ..we cannot describe it clearly.

Response: We have deleted this sentence as you suggested.

7.Line 341 delete  ‘been’ from …most have been failed…

Response: Thank you for your careful proofreading. I'm sorry for making such a basic grammar mistake. We have deleted this word for your suggestion.

Reviewer 2 Report

The manuscript by Nie et al. (biomolecules-1818940) “Research progress of nanoparticles-based CRISPR/Cas9 system for targeted therapy of tumors” reviews some relevant information about nanoparticles-based CRISPR/Cas9 for the application during the therapy of tumors. However, I found some flaws that should be improved for a better version:

Major comments:

-     This manuscript offers a revision of some publications in the field, nevertheless, as can be seen in my comments below, I found this review poor in terms of the analyzed bibliography. Taking into consideration that this manuscript is a revision and the number of existing publications about this topic, I consider that the bibliography section should be more complete, as well as include the appropriate references to the main text.

-     Could the authors carefully revise the whole manuscript for grammatical inconsistencies and correct them to make sentences clearer? Edition and changes in the English language style are necessary.

-     Please, introduce better the term “CRISPR/Cas9” in the introduction. The authors directly described the importance of such a tool but did not explain what it is.

-     Could the authors improve the resolution of Figure 1? Some colors can be hardly seen.

-     I encourage the authors to substitute “we think”, “we believe” and similar for other less speculative comments or discuss them in section “6. Future perspective and challenges”

Specific comments:

·       Line 25: Please, explain further about the genetic landscape and which alterations in DNA sequence.

·       Line 32: could the authors indicate the reference of that research article where CRISPR-Cas9 was firstly used “as a genome editing tool in 2013 in mammalian cells”?

·       Lines 34-40: could the authors add the corresponding references to this text?

·       Lines 41-47: it is not clear to me which references are explaining what the authors described in the main text.

·       Lines 56-57: reference in that sentence?

·       Lines 64-65: please, the reference/s is missed

·       Lines 88-93: references?

·       Lines 102-104: references?

·       Lines 105-115, and 116: what is explained in that text is relevant enough to have only one reference, especially taking into consideration that this manuscript is a revision of other publications.

·       Lines 118-120: could you please rewrite this? Not sure that I could understand well.

·       Lines 123-131: more references are needed

·       Lines 144-148: more references are needed

·       Lines 153-157: the rewriting of this part could significantly improve its understanding for the future readers

·       Lines 338-339: which ones?

·       Lines 364-375: more references are needed

Author Response

Dear Reviewer:
Thank you for your comments concerning our manuscript entitled “Research progress of nanoparticles-based CRISPR/Cas9 system for targeted therapy of tumors.” (ID:1818940). Those comments are all valuable and very helpful for revising and improving our paper. We are really sorry for our shortcomings in the manuscript. We have studied comments carefully and have made correction which we hope meet with approval. Revised portion are marked in red in the paper. The main corrections in the paper and the responds to the reviewer’s comments are as flowing:

1.This manuscript offers a revision of some publications in the field, nevertheless, as can be seen in my comments below, I found this review poor in terms of the analyzed bibliography. Taking into consideration that this manuscript is a revision and the number of existing publications about this topic, I consider that the bibliography section should be more complete, as well as include the appropriate references to the main text.

Response: We agree with your assessment. We have added amounts of references in the manuscript.

2.Could the authors carefully revise the whole manuscript for grammatical inconsistencies and correct them to make sentences clearer? Edition and changes in the English language style are necessary.

Response: We have carefully proofread the grammar of the entire manuscript, and try our best to edited in the English language style.

3.Please, introduce better the term “CRISPR/Cas9” in the introduction. The authors directly described the importance of such a tool but did not explain what it is.

Response: Your comments are very helpful for us. We have revised these suggestions in the introduction.

4.Could the authors improve the resolution of Figure 1? Some colors can be hardly seen.

 Response: We are really sorry for our shortcomings in the figure. We have changed the light yellow to the bright yellow in this figure.

5.I encourage the authors to substitute “we think”, “we believe” and similar for other less speculative comments or discuss them in section “6. Future perspective and challenges”

Specific comments:

Response: This is an interesting perspective. We have revised these comments in the future perspective and challenges.

6.Line 25: Please, explain further about the genetic landscape and which alterations in DNA sequence.

Response: Your comments are very helpful for us. We have detailed the genetic alterations that cause tumorigenesis in this sentence.

7.Line 32: could the authors indicate the reference of that research article where CRISPR-Cas9 was firstly used “as a genome editing tool in 2013 in mammalian cells”?

Lines 34-40: could the authors add the corresponding references to this text?

Lines 41-47: it is not clear to me which references are explaining what the authors described in the main text.

Lines 56-57: reference in that sentence?

Lines 64-65: please, the reference/s is missed

Lines 88-93: references?

Lines 102-104: references?

Lines 105-115, and 116: what is explained in that text is relevant enough to have only one reference, especially taking into consideration that this manuscript is a revision of other publications.

Lines 123-131: more references are needed

Lines 144-148: more references are needed

Lines 338-339: which ones?

Lines 364-375: more references are needed

Response: We apologize for not providing enough references. We have added the related references as you suggested.

8.Lines 118-120: could you please rewrite this? Not sure that I could understand well.

Lines 153-157: the rewriting of this part could significantly improve its understanding for the future readers

Response: Thank you for your careful work. We have rewritten these sentences in the manuscript.

Reviewer 3 Report

The authors in this review focus on a very interesting topic: delivery of the CRISPR/Cas9 system through nanoparticles, a very current and interesting topic for readers. However, it is necessary to work further on the manuscript and make changes, detailed below.

Major changes:

-   Reference 3 (line 27): is a review about BRCA1/2, genes altered in some cancers, but is a review about these genes in cardiovascular diseases, and is not a massive sequence project, please select more appropriate references.  

-          Reference 5 (line 34), maybe authors should use another reference more specific for the phenomenon described

-          Figure 1 and figure 4 are too much similar to figure 1 and figure 7 of the article: Zhan T. et al. 2019 (reference 5); authors should be more original with their figures, please modify it.

-          Lines 41 – 45: Authors should include that low toxicity of CRISPR/Cas9 carriers is also important.

-          Lines 55- 69: I think that authors should explain more about CRISPR/Cas9 system, because is important to understand how it works to describe later how it can be delivered.

-          Many references are missing in the text, and they are very important, especially in a review:

o   Lines 64 – 66

o   Lines 74 – 76

o   Lines 80 – 81

o   Lines 81 – 83

o   Lines 88 – 91

o   Lines 105 – 106

o   Lines 109 – 112

o   Lines 118-119

o   Lines 119-120

o   Lines 134 – 135

o   Lines 174 – 175

o   Lines 175 – 177

o   Lines 181 – 183

o   Lines 183 – 184

o   Lines 189- 190

o   Lines 195 – 196

o   Lines 205 – 207

o   Lines 209 – 210

o   Lines 218 – 221

o   Line 225

o   Lines 225 – 227

o   Lines 243 – 251

o   Lines 253 – 258

o   Lines 267 – 272

o   Lines 277 – 278

o   Lines 287 – 288

o   Lines 288 – 289

o   Lines 305 – 306

o   Lines 364 – 365

o   Line 366

o   Lines 366 – 368

o   Lines 371 – 372

o   Line 372 – 373

o   Line 373 – 375

o   Line 383 – 386

o   Lines 398 – 419

-          Figure 1: The figure caption is too short; authors should explain more and define abbreviations included in the images.

-          Lines 74 – 76: Also transfection commercial agents are used to deliver CRISPR/Cas9 system in the cells, please include it.

-          Lines 81-83: authors should describe more about the multifunctional modification and for which cancer treatment is appropriate

-          Reference 20: I tried to find this reference but it will be available in 2022-10-01, so authors should cite articles that they and readers could access.

-          Lines 163 – 168: authors should explain more about which targeting molecules are used to target the different organs.

-          Lines 174 – 179: Authors said that TME responsive liposomes have attention in the recent years, but they only mention one study, please give more examples about it.

-          Some expressions in English sound estrange, e.g. interactions lead to disintegrate or gather and reduce drug release soon (Line 221); combat chemotherapy resistance (line 297), etc. Please review English in entire manuscript.

-          Please rewrite the following paragraph: Lines 287 (From Epirubicin) – 295

-          Lines 304 – 305: Authors said that antibodies against PD-L1 or PD-1 are used in the treatment of lung cancer, etc. Please be more specific and add references.

-          Lines 316 – 319; explain more this example, how IL-6 protein trap enhance PD-L1 and why in ccrk KO model.

-          Line 325: which therapeutic effects and in which cancer?

-          Line 333: which high curative effects?

-          Rewrite the sentence in lines 336 – 339: it do not sound serious and scientific.

-          Lines 341- 345: information of this paragraph not have sense in this section.

-          Lines 352 – 353: Add references or clinical trial identifiers for the examples.

-          Figure 4: the most important aspect is that it is not a CAR-T treatment, so please correct it. Moreover, explanation of the figure is too short.

-          Lines 368 – 370: it is impossible that CRISPR/Cas9 technology used to deliver CAR into T-cells resulted in tumors, please revise this article and rewrite the information, is difficult to understand.

-          Line 371: “But some treatments for liquid and solid tumors have yet to be found”. Authors should rewrite this sentence, information is not clear and the relation with the paragraph is confuse.

-          Lines 373 – 375: How modification of other genes in t-cells is related with CAR-T cell therapy? Authors should complete more this sentence to be integrated in this paragraph.

-          Line 392 – 393: detail which DNA repair genes are truncated in this study.

-          The authors should do more work and improve the Perspectives and Future Challenges section. The authors focus too much on CAR-T therapy, which is only one section of the review, and should focus on NPs to deliver the CRISPR/Cas9 system. In addition, the authors said that the main limitation of CAR-T is that gene editing cannot be achieved in vivo, but it is not the main limitation, and it is not true that it cannot be used to treat solid tumors, it can but it is more difficult than hematological malignancies, but the reason is not in vivo gene editing, is the problem of T cells reaching the tumor. I believe that the authors do not have enough knowledge about CAR-T to make these claims. Furthermore, it is not the main topic of the review, and therefore the authors should rewrite the section and add references.

Minor changes

-          Some abbreviations are not defined the first time that are mention (e.g. HNC) please, revise and correct it.

-          Authors should mention figures as e.g. Figure 1; instead of Figure.1

-          Line 279: authors wrote, “cite examples”, please delete it.

-          CD8+ is misspelled

-          Line 372: CRISPR/Cas9 instead of CRISPRCas9

Author Response

Dear Reviewer:
Thank you for your comments concerning our manuscript entitled “Research progress of nanoparticles-based CRISPR/Cas9 system for targeted therapy of tumors.” (ID:1818940). Those comments are all valuable and very helpful for revising and improving our paper. We are really sorry for our shortcomings in the manuscript. We have studied comments carefully and have made correction which we hope meet with approval. Revised portion are marked in red in the paper. The main corrections in the paper and the responds to the reviewer’s comments are as flowing:

1.Reference 3 (line 27): is a review about BRCA1/2, genes altered in some cancers, but is a review about these genes in cardiovascular diseases, and is not a massive sequence project, please select more appropriate references.  

Response: Thank you for your careful work. We have added the related references as you suggested.

2.Reference 5 (line 34), maybe authors should use another reference more specific for the phenomenon described

Response: Thank you for your careful work. We have changed this reference as you suggested.

3.Figure 1 and figure 4 are too much similar to figure 1 and figure 7 of the article: Zhan T. et al. 2019 (reference 5); authors should be more original with their figures, please modify it. The figure caption is too short; authors should explain more and define abbreviations included in the images. Figure 4: the most important aspect is that it is not a CAR-T treatment, so please correct it. Moreover, explanation of the figure is too short.

Response: Your comments are very helpful for us. We have repainted two figures in the manuscript.

4.Lines 41 – 45: Authors should include that low toxicity of CRISPR/Cas9 carriers is also important.

Response: This is an interesting perspective. We have added this content in the manuscript.

5.Lines 55- 69: I think that authors should explain more about CRISPR/Cas9 system, because is important to understand how it works to describe later how it can be delivered.

Response: Your comments are very helpful for us. We have revised these suggestions in the introduction.

6.Many references are missing in the text, and they are very important, especially in a review:

Lines 64 – 66

Lines 74 – 76

Lines 80 – 81

Lines 81 – 83

Lines 88 – 91

Lines 105 – 106

Lines 109 – 112

Lines 118-119

Lines 119-120

Lines 134 – 135

Lines 174 – 175

Lines 175 – 177

Lines 181 – 183

Lines 183 – 184

Lines 189- 190

Lines 195 – 196

Lines 205 – 207

Lines 209 – 210

Lines 218 – 221

Line 225

Lines 225 – 227

Lines 243 – 251

Lines 253 – 258

Lines 267 – 272

Lines 277 – 278

Lines 287 – 288

Lines 288 – 289

Lines 305 – 306

Lines 364 – 365

Line 366

Lines 366 – 368

Lines 371 – 372

Line 372 – 373

Line 373 – 375

Line 383 – 386

Lines 398 – 419

 Response: We apologize for not providing enough references. We have added the related references as you suggested.

7.Lines 74 – 76: Also transfection commercial agents are used to deliver CRISPR/Cas9 system in the cells, please include it.

Response: We have added this content in the manuscript as you suggested.

8.Lines 81-83: authors should describe more about the multifunctional modification and for which cancer treatment is appropriate

Response: Your suggestions are very helpful for us. We have added this content in the manuscript.

9.Reference 20: I tried to find this reference but it will be available in 2022-10-01, so authors should cite articles that they and readers could access.

Response: Thank you for your careful work. We have changed this reference as you suggested.

10.Lines 163 – 168: authors should explain more about which targeting molecules are used to target the different organs.

Response: Your comments are very helpful for us. We have added this content in the manuscript.

11.Lines 174 – 179: Authors said that TME responsive liposomes have attention in the recent years, but they only mention one study, please give more examples about it.

Response: We have added the related references as you suggested.

12.Some expressions in English sound estrange, e.g. interactions lead to disintegrate or gather and reduce drug release soon (Line 221); combat chemotherapy resistance (line 297), etc. Please review English in entire manuscript.

Response: Thank you for your careful work. We have rewritten these sentences in the manuscript.

13.Please rewrite the following paragraph: Lines 287 (From Epirubicin) – 295

Response: We have rewritten this paragraph in the manuscript.

14.Lines 304 – 305: Authors said that antibodies against PD-L1 or PD-1 are used in the treatment of lung cancer, etc. Please be more specific and add references.

Response: We have revised these comments in this sentence.

15.Lines 316 – 319; explain more this example, how IL-6 protein trap enhance PD-L1 and why in ccrk KO model.

Response: We have revised these comments in the manuscript.

16.Line 325: which therapeutic effects and in which cancer? Line 333: which high curative effects?

Response: This combined therapy achieved a tumor suppression rate of about 85% in breast tumor xenografts model. We have added this content in this sentence.

17.Rewrite the sentence in lines 336 – 339: it do not sound serious and scientific.

Response: We have rewritten this sentence in the manuscript.

18.Lines 341- 345: information of this paragraph not have sense in this section.

Response: We agree with your assessment. We have deleted this paragraph in the manuscript because of your suggestion.

19.Lines 352 – 353: Add references or clinical trial identifiers for the examples.

Response: We have added the related references as you suggested.

20.Lines 368 – 370: it is impossible that CRISPR/Cas9 technology used to deliver CAR into T-cells resulted in tumors, please revise this article and rewrite the information, is difficult to understand.

Response: We are sorry for our grammatical errors. We have rewritten this sentence in the manuscript.

21.Line 371: “But some treatments for liquid and solid tumors have yet to be found”. Authors should rewrite this sentence, information is not clear and the relation with the paragraph is confuse.

Response: We have rewritten this sentence in the manuscript.

22.Lines 373 – 375: How modification of other genes in t-cells is related with CAR-T cell therapy? Authors should complete more this sentence to be integrated in this paragraph.

Response: This is an interesting perspective. We have revised these comments in the manuscript.

23.Line 392 – 393: detail which DNA repair genes are truncated in this study.

Response: MLH1.

24.The authors should do more work and improve the Perspectives and Future Challenges section. The authors focus too much on CAR-T therapy, which is only one section of the review, and should focus on NPs to deliver the CRISPR/Cas9 system. In addition, the authors said that the main limitation of CAR-T is that gene editing cannot be achieved in vivo, but it is not the main limitation, and it is not true that it cannot be used to treat solid tumors, it can but it is more difficult than hematological malignancies, but the reason is not in vivo gene editing, is the problem of T cells reaching the tumor. I believe that the authors do not have enough knowledge about CAR-T to make these claims. Furthermore, it is not the main topic of the review, and therefore the authors should rewrite the section and add references.

Response: Your comments are very helpful for us. We have added this content in the manuscript.

25.Some abbreviations are not defined the first time that are mention (e.g. HNC) please, revise and correct it.

Response: Thank you for your careful work. We have added this content in the manuscript.

26.Authors should mention figures as e.g. Figure 1; instead of Figure.1

Response: Thank you for your careful work. We have changed this word as you suggested.

27.Line 279: authors wrote, “cite examples”, please delete it.

Response: We have completed the modification for your suggestion.

28.CD8+ is misspelled

Response: Thank you for your careful work. We have changed this word as you suggested.

29.Line 372: CRISPR/Cas9 instead of CRISPRCas9

Response: Thank you for your careful work. We have changed this word as you suggested.

Round 2

Reviewer 3 Report

The authors have made several changes to the first version of the manuscript. However, they still have quite a few concept and reference problems, authors should thoroughly review the article and spend time checking/modifying references.

Major changes:

-          Some references still missing in the text:

o   Lines 25-26

o   Lines 93-94

o   Lines 94-95

o   Lines 95-97 

-          Authors used too many reviews as references, maybe some of them should be changed for the original papers or more specific papers, authors should revise entire manuscript (e.g. below):

o   Reference 22 (lines 111-112)

o   Reference 53 (line 138)

o   Reference 54 (line 139 + 147)

o   Reference 64

o   Reference 124 

-          Some references are incorrectly used, authors should revise entire manuscript (e.g. below):

o   Reference 10 and 11 are not about epigenetic and transcriptional modifications.

o   Reference 12 is not about stability of CRISPR/Cas9 system in serum.

o   Reference 41, please use a more specific reference

o   Reference 50 is not about a plasmid encoding all the CRISPR/Cas9 elements, is an article about a normal plasmid for CRISPR/Cas9 and a sgRNA separately, and authors test its transfection at different time points, and also in vitro and in vivo.

o   Reference 51 is not about a plasmid encoding all the CRISPR/Cas9 elements is a paper using CRISPR/Cas9 RNP.

o   Reference 54, this paper do not explain the off-target effects.

o   Reference 62

o   Reference 64

o   Reference 86: Maybe authors should change for another reference more specific about nanowire NPs.

o   Reference 90: this review is only about gene therapy with NPs, authors should use another reference/s for gene therapy (without NPs), immunotherapy and photothermal therapy; or change the sentence referenced 

-          Some expressions sound strange or not scientific, authors should revise entire manuscript (e.g. below):

o   Line 55: stress

o   Line 123: or so on

o   Line 158 + 246: other shapes; maybe authors could use more scientific words.

o   Line 181: 3D tumor spheroid model instead of 3D tumor ball model

o   Line 217: Transferring, maybe better transfer

o   Line 375: “We encourage interested readers to read other reviews on CRISPR/Cas9”; please remove this sentence.

o   Line 390: “multiplication cultured”, is better to say expanded

-          Authors included the commercial transfection agents (lines 89-90) but is not integrated in the paragraph; please rewrite this part of the paragraph.  

-          Lines 100-101: “researchers have attached importance to NPs good targeting via multifunctional modification”; this sentence is strange and the meaning is not clear, please rewrite the sentence.

-          Lines 103-112: This paragraph is very similar to a paragraph published in another article by other authors (Givens BE, et al. The APPS journal (2018)), please change this paragraph.

-          Figure 2. Authors included an image of mitochondria in the step: CRISPR/Cas9 system was release into the cytoplasm, but this is confusing, the authors should remove it. In addition, figure captions need to be expanded.

-          Lines 134-135: “editing efficiency is between 4% and 36%”, I could not found this information in the article (ref. 52), please check it and if this information is not in the original article, please remove it.

-          Lines 138-139: “Cas9 mRNA can be transiently expressed in the nucleus and eventually completely removed from the nucleus, which is beneficial to reduce off-target effects”. Authors should explain better and be clearer; I understand that they want to explain that if Cas9 is delivered as an mRNA, its expression is transient so its effect in the nucleus is less than if it is integrated as a vector, so it reduces off-targets.  

-          Figure 3: authors should detail more what they show in the figure and include the meaning of the abbreviations, even if these have already been mentioned in the text.

-          Lines 240-241:” Thus, Wang et al. designed a photothermal triggered gold NPs for systematic delivery of 240 CRISPR/Cas9[60]”; authors should mention that these NPs contain gold but are Lipid-Encapsulated Gold Nanoparticles.

-          Lines 241 – 245: Authors should mention that magnetic lipoplexes were used in porcine cells. Are any studies on human cells?

-          Line 247: “polymer chain codes stated al nano polymer gel”, authors have to rewrite this sentence, the meaning is not clear.

-          Line 249: if nanogels reduced CRISPR/Cas9 system release, why is interesting this method to deliver CRISPR/Cas9 system? Explain it better.

-          Line 273: “chemotherapy, radiotherapy, surgery and etc.”; if authors only mention the most common ones, they should mention it and not write “and etc.”

-          Lines 302-305: Example showed by authors is an example of the use of CRISPR/Cas9 to modify an oncogene, not a tumor suppressor, despite the fact that this causes a restoration of p53.

-          Line 325: which therapeutic effects and in which cancer?

-          Line 337, references 101-103: two of them are from 2016 and another from 2019; maybe authors should include clinical trials more up-to-date (e.g. references 109-112) because anti-PD-1 and anti-PD-L1 treatments are improved over time.

-          Line 337: Authors said that only a minority of patients presented immune responses in clinical trials with PD-1 and PD-L1; Are you sure? Authors should revise bibliography and clinical trials.

-          Lines 384-385: Authors mentioned other clinical trials where applied the same concept of PD-1 knockout, but is not totally true, they are articles about anti-PD-1 clinical trials, which is not relevant for this review.

-          Authors improved section Future perspective and challenges, but from my point of view, the authors should work on it a little more, relate concepts, integrate the information and give an overview of what is shown in the article. Currently it is more a summary of each section, without relation.

Minor changes

-          Section “CRISPR/Cas9 system” (lines 60-79) need paragraph alignment (Justify)

-          Line 322: Please change SAS cells for SAS cell line.

-          Figure captions of Figure 4 need paragraph alignment (Justify)

-          Line 402: CAR-T instead of CAR T (following the same abbreviation in all manuscript)

Author Response

Dear Reviewer:
Thank you for your comments concerning our manuscript entitled “Research progress of nanoparticles-based CRISPR/Cas9 system for targeted therapy of tumors.” (ID:1818940). Those comments are all valuable and very helpful for revising and improving our paper. We are really sorry for our shortcomings in the manuscript. We have studied comments carefully and have made correction which we hope meet with approval. Revised portion are marked in red in the paper. The main corrections in the paper and the responds to the reviewer’s comments are as flowing:

Major changes:

  1. Some references still missing in the text:

Lines 25-26

Lines 93-94

Lines 94-95

Lines 95-97 

Response: We apologize for not providing enough references. We have added the related references as you suggested.

  1. Authors used too many reviews as references, maybe some of them should be changed for the original papers or more specific papers, authors should revise entire manuscript (e.g. below):

Reference 22 (lines 111-112)

Reference 53 (line 138)

Reference 54 (line 139 + 147)

Reference 64

Reference 124 

Response: We have cited the original papers as our references as you suggested.

  1. Some references are incorrectly used, authors should revise entire manuscript (e.g. below):

Reference 10 and 11 are not about epigenetic and transcriptional modifications.

Reference 12 is not about stability of CRISPR/Cas9 system in serum.

Reference 41, please use a more specific reference

Reference 50 is not about a plasmid encoding all the CRISPR/Cas9 elements, is an article about a normal plasmid for CRISPR/Cas9 and a sgRNA separately, and authors test its transfection at different time points, and also in vitro and in vivo.

Reference 51 is not about a plasmid encoding all the CRISPR/Cas9 elements is a paper using CRISPR/Cas9 RNP.

Reference 54, this paper do not explain the off-target effects.

Reference 62

Reference 64

Reference 86: Maybe authors should change for another reference more specific about nanowire NPs.

Reference 90: this review is only about gene therapy with NPs, authors should use another reference/s for gene therapy (without NPs), immunotherapy and photothermal therapy; or change the sentence referenced 

Response: We are sorry for using incorrect references because of our carelessness. We have changed the related references as you suggested. We did not change reference 54, because authors introduced genome editing reagents into single-cell bovine embryos to compare the effect of Cas9 mRNA and protein on the mutation efficiency and evaluated potential off-target mutations utilizing next generation sequencing, and found injected with Cas9 mRNA with little to no unintended off-target mutations detected compared to those injected with Cas9 protein in this paper, which could prove our points: Cas9 mRNA is beneficial to reduce off-target effects.

  1. Some expressions sound strange or not scientific, authors should revise entire manuscript (e.g. below):

Line 55: stress

Line 123: or so on

Line 158 + 246: other shapes; maybe authors could use more scientific words.

Line 181: 3D tumor spheroid model instead of 3D tumor ball model

Line 217: Transferring, maybe better transfer

Line 375: “We encourage interested readers to read other reviews on CRISPR/Cas9”; please remove this sentence.

Line 390: “multiplication cultured”, is better to say expanded

Response: Thank you for your careful work. We have revised these parts in the manuscript as you suggested.

5.Authors included the commercial transfection agents (lines 89-90) but is not integrated in the paragraph; please rewrite this part of the paragraph.  

Response: We have rewritten this paragraph in the manuscript.

6.Lines 100-101: “researchers have attached importance to NPs good targeting via multifunctional modification”; this sentence is strange and the meaning is not clear, please rewrite the sentence.

Response: We have rewritten this sentence in the manuscript.

7.Lines 103-112: This paragraph is very similar to a paragraph published in another article by other authors (Givens BE, et al. The APPS journal (2018)), please change this paragraph.

Response: We have rewritten this paragraph in the manuscript.

8.Figure 2. Authors included an image of mitochondria in the step: CRISPR/Cas9 system was release into the cytoplasm, but this is confusing, the authors should remove it. In addition, figure captions need to be expanded.

Response: We have deleted this content and added figure captions as you suggested.

9.Lines 134-135: “editing efficiency is between 4% and 36%”, I could not found this information in the article (ref. 52), please check it and if this information is not in the original article, please remove it.

Response: We evaluated the editing efficiency is between 4% and 36% according to figure 2B in this original article.

10.Lines 138-139: “Cas9 mRNA can be transiently expressed in the nucleus and eventually completely removed from the nucleus, which is beneficial to reduce off-target effects”. Authors should explain better and be clearer; I understand that they want to explain that if Cas9 is delivered as an mRNA, its expression is transient so its effect in the nucleus is less than if it is integrated as a vector, so it reduces off-targets.  

Response: We have rewritten this sentence in the manuscript.

11.Figure 3: authors should detail more what they show in the figure and include the meaning of the abbreviations, even if these have already been mentioned in the text.

Response: We have repainted this picture in the manuscript.

12.Lines 240-241:” Thus, Wang et al. designed a photothermal triggered gold NPs for systematic delivery of 240 CRISPR/Cas9[60]”; authors should mention that these NPs contain gold but are Lipid-Encapsulated Gold Nanoparticles.

Response: Thank you for your careful work. We have added this content in this sentence as you suggested.

13.Lines 241 – 245: Authors should mention that magnetic lipoplexes were used in porcine cells. Are any studies on human cells?

Response: We have deleted this content and added new research on human cells as you suggested.

14.Line 247: “polymer chain codes stated al nano polymer gel”, authors have to rewrite this sentence, the meaning is not clear.

Response: We have rewritten this sentence in the manuscript.

15.Line 249: if nanogels reduced CRISPR/Cas9 system release, why is interesting this method to deliver CRISPR/Cas9 system? Explain it better.

Response: Nanogels can reduce CRISPR/Cas9 system release in the blood to improve its bioavailability in tumor tissues. We have rewritten this sentence in the manuscript.

16.Line 273: “chemotherapy, radiotherapy, surgery and etc.”; if authors only mention the most common ones, they should mention it and not write “and etc.”

Response: We have deleted “and etc.” in the manuscript.

17.Lines 302-305: Example showed by authors is an example of the use of CRISPR/Cas9 to modify an oncogene, not a tumor suppressor, despite the fact that this causes a restoration of p53.

Response: CRISPR/Cas9 is mostly used to knock out genes, and the knockout of tumor suppressor genes will lead to the formation of tumors, so there is no example of directly applying CRISPR/Cas9 on tumor suppressor genes to treat cancer.

18.Line 325: which therapeutic effects and in which cancer?

Response: Knockout of HuR using CRISPR/Cas9 NPs significantly inhibited tumor growth and improved the survival percent in epirubicin-treated SAS tumor-bearing mice in head and neck cancer.

19.Line 337, references 101-103: two of them are from 2016 and another from 2019; maybe authors should include clinical trials more up-to-date (e.g. references 109-112) because anti-PD-1 and anti-PD-L1 treatments are improved over time.

Response: We have changed the related references as you suggested.

20.Line 337: Authors said that only a minority of patients presented immune responses in clinical trials with PD-1 and PD-L1; Are you sure? Authors should revise bibliography and clinical trials.

Response: We have deleted this content in the manuscript.

21.Lines 384-385: Authors mentioned other clinical trials where applied the same concept of PD-1 knockout, but is not totally true, they are articles about anti-PD-1 clinical trials, which is not relevant for this review.

Response: We have deleted this content as you suggested.

22.Authors improved section Future perspective and challenges, but from my point of view, the authors should work on it a little more, relate concepts, integrate the information and give an overview of what is shown in the article. Currently it is more a summary of each section, without relation.

 Response: We have rewritten this paragraph in the manuscript.

23.Minor changes

Section “CRISPR/Cas9 system” (lines 60-79) need paragraph alignment (Justify)

Line 322: Please change SAS cells for SAS cell line.

Figure captions of Figure 4 need paragraph alignment (Justify)

Line 402: CAR-T instead of CAR T (following the same abbreviation in all manuscript)

Response: Thank you for your careful work. We have revised these parts in the manuscript as you suggested.